# Highly Enhanced Photocatalytic Performances of Composites Consisting of Silver Phosphate and N-Doped Carbon Nanomesh for Oxytetracycline Degradation

**DOI:** 10.3390/ijerph192214865

**Published:** 2022-11-11

**Authors:** Shehua Tong, Zhibing Liu, Yan Lin, Chunping Yang

**Affiliations:** 1College of Environmental Science and Engineering, Hunan University and Key Laboratory of Environmental Biology and Pollution Control (Hunan University), Ministry of Education, Changsha 410082, Hunan, China; 2Guangdong Provincial Key Laboratory of Petrochemical Pollution Processes and Control, Key Laboratory of Petrochemical Pollution Control of Guangdong Higher Education Institutes, School of Environmental Science and Engineering, Guangdong University of Petrochemical Technology, Maoming 525000, Guangdong, China; 3School of Environmental and Chemical Engineering, Nanchang Hangkong University, Nanchang 330063, Jiangxi, China

**Keywords:** visible light catalyst, carbon material, silver phosphate, oxytetracycline

## Abstract

Photocatalytic technology based on silver phosphate (Ag_3_PO_4_) has excellent potential in removing antibiotic pollutants, but the low separation rate of photogenerated hole-electron pairs restricts the application of the photocatalyst. In this study, it was found that the combination of nitrogen-doped carbon (NDC) with carbon defects and Ag_3_PO_4_ can significantly enhance the photocatalytic ability of Ag_3_PO_4_. After it was exposed to visible light for 5 min, the photocatalytic degradation efficiency of oxytetracycline (OTC) by the composite photocatalyst Ag_3_PO_4_@NDC could reach 100%. In addition, the structure of NDC, Ag_3_PO_4,_ and Ag_3_PO_4_@NDC was systematically characterized by SEM, TEM, XRD, Raman, and EPR. The XPS results revealed intense interface interaction between Ag_3_PO_4_ and NDC, and electrons would transfer from Ag_3_PO_4_ to the NDC surface. A possible mechanism for enhancing the photocatalytic reaction of the Ag_3_PO_4_@NDC composite catalyst was proposed. This study provides a highly efficient visible light catalytic material, which can be a valuable reference for designing and developing a new highly efficient visible light catalyst.

## 1. Introduction

Antibiotics are widely used in the healthcare and agricultural sectors for disease prevention and treatment [1,2]. There are many kinds of antibiotics, and they are used in large quantities. Due to the epidemic situation, the use of antibiotics is growing [3,4]. According to statistics, the global consumption of antibiotics in 2018 was 4.02 billion tons, and the use of antibiotics increased by 46% from 2000 to 2018 [5]. Antibiotics administered to humans and animals are often unable to be completely absorbed and excreted as is or in the form of metabolites. Therefore, antibiotics or their metabolic intermediates are often detected in the urine and feces of drug-applying organisms [6,7]. Antibiotics released into the environment will change the antibiotic resistance of microorganisms [8,9], trigger the spread of antibiotic resistance genes [10], and pose great environmental and ecological risks.

Photocatalytic technology is an environmentally friendly pollution control technology and has great potential to remove antibiotics [11,12]. The key to this technology is to improve the photocatalytic performance and stability of the photocatalyst [13,14]. As a typical visible light-responsive photocatalyst, silver phosphate (Ag_3_PO_4_) has been widely studied for its excellent photocatalytic oxidation capacity [15,16]. However, the low photostability and electron-hole separation efficiency affect the practical application of Ag_3_PO_4_ [17]. The composite photocatalyst of Ag_3_PO_4_ and other materials can further improve its optical absorption performance and charge separation efficiency [18]. Therefore, most of the researchers have made various attempts to overcome these defects, for example, morphology modification [19], element doping [20], crystal plane engineering [21], designing of heterostructures [22], and addition of carbon materials [23]. The addition of carbon materials can promote the separation and transport of photogenerated electron-hole pairs, and the high specific surface area of carbon materials can promote the adsorption process of pollutants [24,25]. For example, Hu et al. [26] have prepared Ag_3_PO_4_/TiO_2_/C photocatalyst and found that the stability and photocatalytic performance of Ag_3_PO_4_ have been greatly improved, and the composite catalyst has a narrower band gap and a wider light absorption range. The presence of carbon materials improves its electron transfer ability and cycle stability, thereby reducing the recombination of photogenerated hole-electron pairs. In addition, the presence of carbon materials can effectively reduce the reduction of silver ions and keep the crystal structure of Ag_3_PO_4_ intact. Lai et al. found that reasonable design and construction of carbon defects coupled with heteroatom doping can more effectively control the charge distribution of carbon materials, which has greater advantages for the construction of an internal electric field at the interface of the composite photocatalyst. In addition, the presence of carbon materials can effectively reduce the reduction of silver ions and keep the crystal structure of Ag_3_PO_4_ intact [27,28]. However, as far as we know, there is no relevant report on combining Ag_3_PO_4_ with carbon materials containing carbon defects and heteroatom doping.

Therefore, in this study, we first synthesized N-doped two-dimensional carbon nanomesh (NDC) with carbon defects and compounded it with Ag_3_PO_4_ to promote the separation and migration of interface charges, thus further improving the catalytic performance of pure Ag_3_PO_4_. Under visible light conditions, oxytetracycline was used as the target pollutant to study the photocatalytic activity of the catalyst, evaluate the catalytic degradation performance for pollutants, and explore its catalytic mechanism. The results of this paper are of great significance to developing and preparing new photocatalytic materials for the efficient degradation of organic pollutants.

## 2. Materials and Methods

### 2.1. Experimental Materials

Reagents such as disodium hydrogen phosphate hydrate (Na_2_HPO_4_·12H_2_O), silver nitrate (AgNO_3_), and oxytetracycline (OTC) were purchased from Sinopharm Chemical Reagent Co., Ltd. (Shanghai, China). All reagents and materials were analytically pure, and deionized water with a resistivity of 18.25 MΩ·cm was used.

### 2.2. Preparation of Ag_3_PO_4_

In the preparation process of this study, Ag_3_PO_4_ was prepared by reacting Na_2_HPO_4_·12H_2_O with AgNO_3_. The preparation of the Ag_3_PO_4_ sample is available in the Appendix A.

### 2.3. Preparation of N-Doped Carbon Defects (NDC)

The synthesis of NDC referred to the synthesis method of Lai et al. [28]. First, uniformly mix urea (1.125 g) and p-phenylene diisocyanate (3.0 g) in tetrahydrofuran (150 mL), disperse them by ultrasound for 10 min, and then stir the mixed solution at room temperature for 12 h to achieve complete polymerization. The material was obtained by suction filtration and washing with tetrahydrofuran, and the precursor was obtained by vacuum drying at 60 °C for 12 h. Put a proper amount of the precursor into a quartz boat with a cover, and then transfer it to a tubular furnace. After the N_2_ atmosphere was turned on for 30 min, start the temperature rise program of the tubular furnace, and increase the heating rate to 600 °C at 2 °C/min for 3 h, then increase the heating rate to 1000 °C at 5 °C/min for 1 h, and take it out after natural cooling to room temperature under the N_2_ atmosphere to obtain NDC materials.

### 2.4. Synthesis of Ag_3_PO_4_@NDC

The synthesis of Ag_3_PO_4_@NDC adopted the electrostatic self-assembly method, referring to Yang et al. [29]. The preparation method of the Ag_3_PO_4_@NDC sample was given in detail in the Appendix A.

### 2.5. Characterization

The morphologies of the samples were determined by s Field emission scanning electron microscopy (FESEM, Hitachi Regulus8100, Tokyo, Japan) and transmission electron microscopy (TEM, TecnaiG2 F20, FEI, Hillsboro, OR, USA). The chemical composition of the samples was analyzed by X-ray photoelectron spectroscopy (XPS, ESCALAB 250Xi, Thermo Fisher, Waltham, MA, USA) and energy dispersive X-ray spectroscopy (EDS, Hitachi, Tokyo, Japan). Raman spectra were obtained on a confocal micro Raman spectrometer (Horiba Jobin Yvon LabRAM HR800, Paris, France) under 633 nm laser excitation. The crystal structures of the prepared samples were characterized by an X-ray diffractometer (Bruker AXS D8 Advances, Karlsruhe, Germany) with Cu-Ka radiation (λ = 0.15406 nm). The UV-vis diffused reflectance spectra (UV-vis DRS) were obtained by UV-vis-NIR spectrophotometer (U-4100, Hitachi, Tokyo, Japan). The photocurrent response curve of the prepared samples was measured by a three-electrode cell in 0.5 mol/L Na_2_SO_4_ aqueous solutions and a CHI 760E workstation (CH Instruments, Shanghai, China). The electron spin resonance (ESR) signals of radicals spin-trapped by spin-trapped reagent 5, 5-dimethyl-l-pyrroline N-oxide (DMPO) were examined on a JES FA200 electron paramagnetic resonance spectrometer (JEOL, Tokyo, Japan) under visible light irradiation (λ > 420 nm).

### 2.6. Photocatalysis Experiment

The degradation of oxytetracycline as a target pollutant under visible light irradiation was used to evaluate the photocatalytic performance of the obtained samples. A 300 W xenon lamp with a 420 nm cut-off filter was used as the light source. By circulating water, the temperature during the whole experiment was kept consistent with the indoor temperature. 50 mg photocatalyst was put and diffused in the oxytetracycline aqueous solution (100 mL, 20 mg/L). The mixed solution was stirred for 30 min in the dark to achieve adsorption-desorption equilibrium, and then the solution was irradiated with a 300 W xenon lamp. 1 mL of suspension was taken out at certain intervals and filtered using a 0.22 μm microporous filter to separate particles. The concentration of oxytetracycline after the reaction was determined using ultraviolet detection and Agilent ZORBAX SB-C18 (5 μm × 4.6 mm × 250 mm) reversed-phase column. The detectors of OTC were respectively set at the wavelength of 280 nm. The test parameters of the OTC in samples were available in the Appendix A.

## 3. Results and Discussion

### 3.1. Characterization of Catalyst

The morphology and microstructure of the samples were studied by SEM. As demonstrated in Figure 1a, the precursor PU had a two-dimensional blocky structure. In the process of annealing PU precursor to form NDC, PU spontaneously self-assembled into two-dimensional (2D) thin nanosheets with rough surface structure. The width of the nanosheets was about 500 nm, and the length was 5–20 μm. Moreover, the carbon nanosheets were composed of nanotubes (Figure 1b,c), which might result from the interaction of the H bond in the self-assembled PU molecular structure [28]. The Ag_3_PO_4_ monomer had a polyhedral structure, mainly shown by the small particle size of the monomer particles attached to the large particle crystal (Figure 1d). In Figure 1e,f, it could be observed that the Ag_3_PO_4_ crystal was closely combined with a sheet and tubular NDC. The Ag_3_PO_4_ particles were in close contact with the NDC surface, mainly due to the electrostatic attraction between Ag^+^ and NDC (with negative surface charge) [30]. SEM results showed that Ag_3_PO_4_@NDC composite catalyst had been successfully synthesized and had good interface contact.

To determine the microstructure and element composition, the TEM (TecnaiG2 F20, FEI) images of the photocatalyst were tested. Figure 2a–c was the TEM images of NDC at different magnifications. As shown in Figure 2a, NDC is a nanosheet with a two-dimensional structure composed of nanotubes. From Figure 2b,c, it could be found that with the increase of magnification, there were uniform pore structures on the carbon material due to the carbon defects caused by the removal of uniformly distributed urea units on the aromatic polymer chain of PU after annealing. Figure 1e and Figure 2d show that NDC was attached to or closely adhered to Ag_3_PO_4_ particles. EDS analysis and cross-section element mapping test were carried out for Ag_3_PO_4_@NDC composite catalyst, as shown in Figure 3. The elements of Ag, P, O, C and N could be distinctly observed, indicating that Ag_3_PO_4_ had a good contact surface with NDC. This might be due to the electrostatic attraction between Ag^+^ and NDC (with a negative charge on the surface) during the preparation process, so Ag_3_PO_4_ particles grew along the surface of NDC carbon nanonets (Figure 2d), further indicating that Ag_3_PO_4_ and NDC were successfully compounded and had good contact surfaces.

In order to understand the crystal structure of the photocatalysts, the prepared samples were measured by XRD spectroscopy, as shown in Figure 4. After being compounded with NDC, the crystal structure of Ag_3_PO_4_ was intact, and it was shown as a body-centered cubic system Ag_3_PO_4_ (JCPDS No. 060505) [31]. The silver phosphate crystal facet exposure could be obtained by comparing the peak intensity of crystal planes. Although the positions of the diffraction peaks of Ag_3_PO_4_ and Ag_3_PO_4_@NDC were almost the same, the intensity ratio of the diffraction peaks of Ag_3_PO_4_ in different samples had obvious changes. It was reported that the (222) crystal facet of Ag_3_PO_4_ had the highest surface energy compared with the (200) and (110) crystal facets, and higher surface energy could further enhance the photocatalytic activity [32]. The intensity ratios of the (222) and (200) diffraction peaks of Ag_3_PO_4_ and Ag_3_PO_4_@NDC were 0.95 and 1.27, respectively. This result showed that the addition of NDC affected the crystal surface exposure of Ag_3_PO_4_, which could cause the highly active crystal surface exposure of silver phosphate, further promoting the provision of its photocatalytic activity.

In order to test the light absorption performance of the samples, UV-Vis DRS characterization of the photocatalyst was conducted. The results are shown in Figure 4b. The spectral results showed that Ag_3_PO_4_ and Ag_3_PO_4_@NDC had good optical absorption performance when the wavelength was below 500 nm. However, compared with the Ag_3_PO_4_ monomer, Ag_3_PO_4_@NDC had higher light absorption intensity in the wavelength of 500~800 nm, which indicated that NDC had successfully compounded with Ag_3_PO_4_ and improved the light absorption ability of the catalyst.

The band gap width (*Eg*) of the photocatalyst could be determined by the Kubelka-Munk equation:(1)$${\left(\alpha h\nu \right)}^{n}=A\left(h\nu -Eg\right)$$

Among the value of *n* is related to the optical transition pattern of the semiconductor (*n* = 2 is the direct transition, *n* = 1/2 is the indirect transition). It could be seen from the fitting results in Figure 4c that the *Eg* of Ag_3_PO_4_ was 2.4 eV, which was easily excited by visible light.

In order to test the carrier separation efficiency of the catalyst sample, the photocurrent response curve was tested. The results are shown in Figure 4d. The figure shows that the sample’s transient photocurrent response showed a repeatable and relatively stable photocurrent distribution in a continuous switching light source cycle. Ag_3_PO_4_@NDC composite had higher photocurrent density than Ag_3_PO_4_ monomer under the same conditions, indicating that the introduction of NDC significantly promoted the separation of photogenerated charge in the photocatalytic reaction system of the composite catalyst and produced more effective photogenerated carriers.

The changes of PU in the formation of NDC were investigated by Raman spectroscopy. Raman spectroscopy showed that the structure had changed during the transformation from PU precursor to NDC. This was because, in the PU molecular skeleton, the stable and periodically arranged benzene rings made the carbon network of PU precursor morphology stable, while the unstable urea unit decomposed to form a rich porous structure. The decomposed urea unit would generate many N-containing fragments, forming more activated carbon edge defects, which could promote efficient N doping into the carbon matrix. The loss of N configuration caused by the decomposition of urea during the formation of NDC in PU could improve the graphitization degree of NDC materials [28]. As shown in Figure 5a, the Raman spectrum of the NDC material prepared showed obvious D-band (1366 cm^−1^) and G-band (1586 cm^−1^), which respectively corresponded to the A1g symmetric respiration mode from a large number of structural defects and the in-plane E_2g_ symmetric vibration from ordered sp^2^ bonded graphite carbon [33]. In the Raman spectrum, two characteristic peaks appeared at 1366 cm^−1^ (D-band) and 1586 cm^−1^ (G-band). D-band and G-band were related to amorphous carbon with structural defects (sp^3^) and graphite carbon (sp^2^). In order to further determine the defect structure formed after annealing and calcination, low-temperature EPR tests were carried out on NDC samples. The results are shown in Figure 5b. NDC had a strong vacancy defect signal peak. This was consistent with the test results obtained by SEM and Raman.

In order to understand the elemental chemical state of Ag_3_PO_4_ and NDC, as well as their interaction and electron transfer, the composite catalysts of Ag_3_PO_4_, NDC and Ag_3_PO_4_@NDC were analyzed by X-ray photoelectron spectroscopy (XPS). As shown in Figure 6a, the binding energies of the two characteristic peaks were 396.64 eV and 399.52 eV, respectively, which were attributed to -N= and-NH-N [34], indicating that N was well doped into carbon materials. The XPS spectrum of Ag_3_PO_4_@NDC shows the characteristic peaks of all elements, such as Ag, P, O, C and N, indicating that Ag_3_PO_4_ was coupled with NDC, which was mutually confirmed by EDS analysis. As shown in Figure 6d, the atlas of Ag 3d showed two peaks: Ag 3d_5/2_ and Ag 3d_3/2_, with binding energies of 367.05 eV and 373.05 eV, respectively, which indicated that the silver element in Ag_3_PO_4_ existed in the form of silver ions [35]. As shown in Figure 6e, the O1s spectrum could be divided into two characteristic peaks. The characteristic peaks at 530.4 eV and 531.9 eV were bulk lattice O of Ag_3_PO_4_ and surface O of catalysts, respectively [32]. As shown in Figure 6f, the characteristic peak at the binding energy of 131.9 eV belonged to the phosphorus element in PO_4_^3−^ [36,37]. Compared with the silver phosphate monomer, the binding energy positions of Ag and O in the composite shifted in the positive direction (Figure 6d,e). Since the binding energy decreases with the surface electron density increase, it meant that the contact between Ag_3_PO_4_ and NDC transferred electrons from Ag_3_PO_4_ to the NDC surface [38,39,40], indicating that there was intense interface interaction and electron transfer in the composite catalyst.

### 3.2. Photocatalytic Degradation Performance of Catalysts

The antibiotic oxytetracycline (OTC) was selected to assess the photocatalytic activity of the above photocatalysts. In every degradation reaction, 50 mg of the photocatalyst was added into OTC (20 mg/L, 100 mL) aqueous solution, and then the mixed solution was stirred in the dark for 30 min to achieve adsorption-desorption equilibrium. Finally, a 300 W xenon lamp (λ > 420 nm) was used as an artificial light source for photodegradation experiments. As shown in Figure 7a, the Ag_3_PO_4_@NDC complex showed higher photocatalytic activity than the monomer Ag_3_PO_4_. After 5 min of visible light irradiation, OTC was 100% degraded by Ag_3_PO_4_ composite loaded with 2.5 mL, 5 mL, 10 mL, and 15 mL NDC. In contrast, the material with 5.0 mL NDC showed the most obvious photocatalytic efficiency, and 100% degradation of OTC was completed within 3 min. However, the degradation efficiency of silver phosphate monomer for OTC could not reach 100% even after 5 min of photo-reaction.

In order to further explore the catalytic activity and degradation performance of composite catalysts with different NDC ratios, the degradation data of Ag_3_PO_4_ monomer and its composite catalyst were simulated by quasi first order kinetic model, respectively. The results are shown in Figure 7b. The Ag_3_PO_4_@5.0 mL NDC composite catalyst had the largest reaction rate constant (K = 1.60 min^−1^), 2.7 times that of the unmodified Ag_3_PO_4_ monomer (0.60 min^−1^). The experimental results showed that the combination of Ag_3_PO_4_ and NDC effectively enhanced the activity and degradation rate of the Ag_3_PO_4_ catalyst. This might be because the addition of NDC could build a built-in electric field at the interface of the composite catalyst, making the electron migration speed more rapid and stable. As an electron potential well, the photogenerated electrons generated by silver phosphate could quickly migrate to the surface of NDC, reducing the internal recombination probability of photogenerated electrons and holes, thus improving the catalytic performance of the photocatalyst.

The free radical quenching experiment explored the main active substances produced by Ag_3_PO_4_@NDC composites in the photoreaction process. Benzoquinone (BQ), disodium ethylenediaminetetraacetic acid (EDTA-2Na), isopropanol (IPA), and silver nitrate (AgNO_3_) were used as scavengers to scavenge superoxide radical (**·**O_2_^−^), photogenerated hole (h^+^), hydroxyl free (**·**OH) and photogenerated electron (e^−^), respectively [41,42,43]. As shown in Figure 8, the addition of IPA had little effect on the degradation and removal of OTC, demonstrating that less **·**OH was involved in the catalytic reaction process. Nevertheless, adding BQ, AgNO_3_ and EDTA-2Na led to a significant decrease in the activity of Ag_3_PO_4_@NDC photocatalyst and its catalytic degradation performance, indicating that **·**O_2_^−^, e^−^ and h^+^ took a leading part in OTC degradation. In particular, the addition of EDTA-2Na almost led to the complete deactivation of the photocatalyst, which indicated that the photogenerated holes played a crucial role in the photocatalytic reaction system. This was related to the intrinsic deep valence band position structure of silver phosphate, which determined that the photogenerated holes on the VB of silver phosphate-based photocatalyst had strong oxidation performance and could directly attack pollutants. In addition, ESR was used to analyze the spin adducts formed after DMPO capture radicals were added to further verify the free radicals generated in the reaction process. As shown in Figure 9a,b, no signal peaks of DMPO-**·**O_2_^−^ and DMPO-**·**OH were found under dark conditions, which proved that **·**O_2_^−^ and **·**OH free radicals were produced in the process of photocatalysis under light conditions. Under visible light conditions, the signal peaks of DMPO-**·**O_2_^−^ and DMPO-**·**OH were visible, and the peak intensity was significantly increased after 5 to 10 min of irradiation, which further confirmed that the surface of Ag_3_PO_4_@NDC catalyst could successfully produce **·**O_2_^−^ and **·**OH radicals under visible light irradiation.

### 3.3. Reaction Mechanism of Ag_3_PO_4_@NDC

According to the above experiment, a possible photocatalytic mechanism of Ag_3_PO_4_@NDC material was proposed. As shown in Figure 10, under visible light, the electrons of Ag_3_PO_4_ transition from the valence band (VB) to the conduction band (CB) and form holes in the VB. The electrons quickly migrated to the surface of NDC and then reacted with water and oxygen to generate **·**O_2_^−^ and **·**OH, thus oxidizing organic pollutants. At the same time, the photogenerated holes on VB had strong oxidation capacity, which could directly attack, oxidize and decompose pollutants. The remarkable enhancement of the photocatalytic activity of Ag_3_PO_4_@NDC could be attributed to the following reasons. Firstly, the introduction of NDC made Ag_3_PO_4_ have stronger light absorption performance, which could generate more photogenerated electrons and holes. Secondly, the addition of NDC could promote the formation of an internal electric field between the interface of Ag_3_PO_4_ and NDC and promote the rapid migration and separation of photogenerated carriers. Thirdly, the two-dimensional nanosheet structure with defects provided more reactive sites for the catalytic reaction and promoted the surface reaction process of photocatalysis.

## 4. Conclusions

In this paper, Ag_3_PO_4_@NDC composites were successfully prepared by electrostatic self-assembly of N-doped carbon (NDC) with defect structure and Ag_3_PO_4_. The prepared Ag_3_PO_4_@NDC composite photocatalyst material had excellent visible light catalytic activity. Under the condition of visible light irradiation for 5 min, the catalytic degradation efficiency of OTC could reach 100%, and the photocatalytic reaction rate was 2.7 times that of silver phosphate monomer. The free radical capture experiment and electron spin resonance test showed that the main active substances in the Ag_3_PO_4_@NDC photocatalytic reaction were O_2_^−^, e^−^, and h^+^. The significantly enhanced photocatalytic activity was mainly due to the intense interface interaction and electron transfer process at the interface between Ag_3_PO_4_ and NDC. This experiment provides an efficient photocatalyst material for removing antibiotics and has reference significance for designing a new efficient visible light catalyst.

## Figures and Tables

**Figure 1 ijerph-19-14865-f001:**
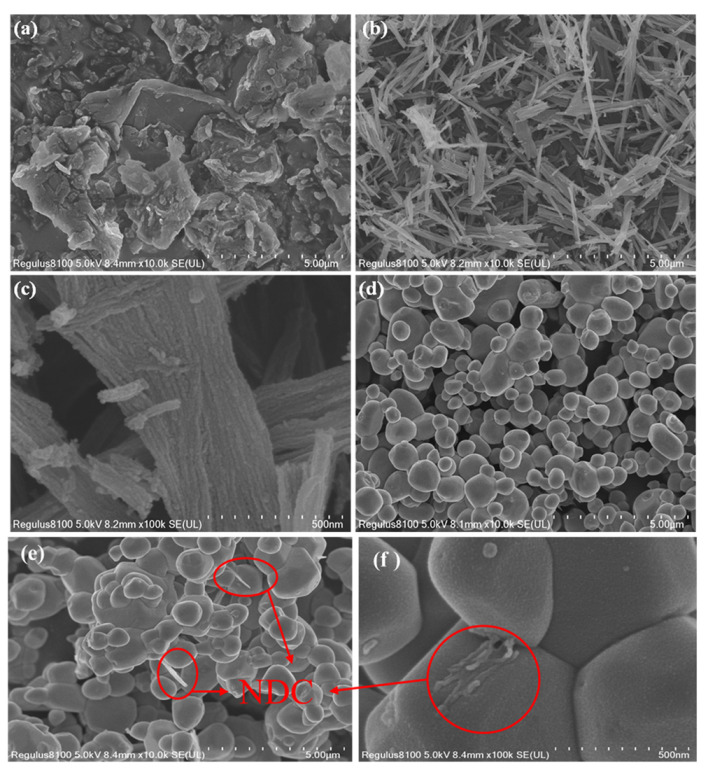
SEM images of (**a**) PU, (**b**,**c**) NDC, (**d**) Ag_3_PO_4_ and (**e**,**f**) Ag_3_PO_4_@NDC.

**Figure 2 ijerph-19-14865-f002:**
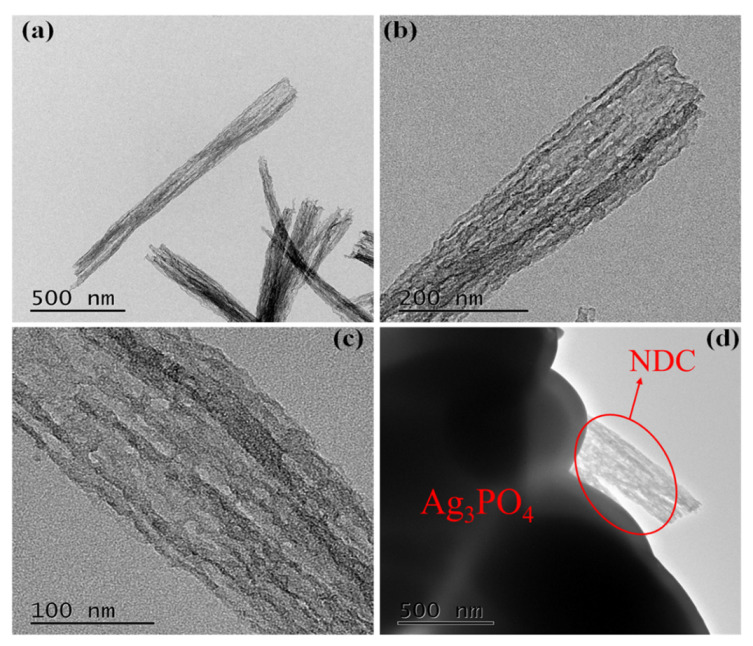
TEM images of NDC at different magnifications (**a**–**c**) and TEM image of Ag_3_PO_4_@NDC (**d**).

**Figure 3 ijerph-19-14865-f003:**
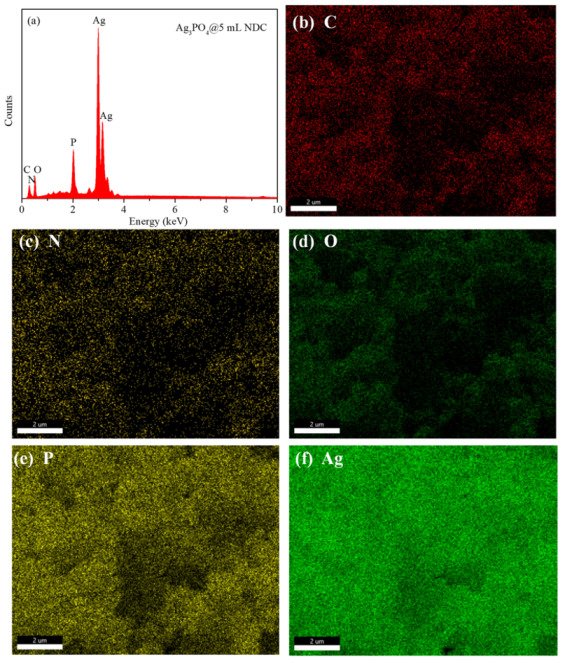
SEM-EDS elemental mapping images of Ag_3_PO_4_@NDC composite: (**a**) EDS spectrum and distribution of (**b**) C, (**c**) N, (**d**) O, (**e**) P and (**f**) Ag.

**Figure 4 ijerph-19-14865-f004:**
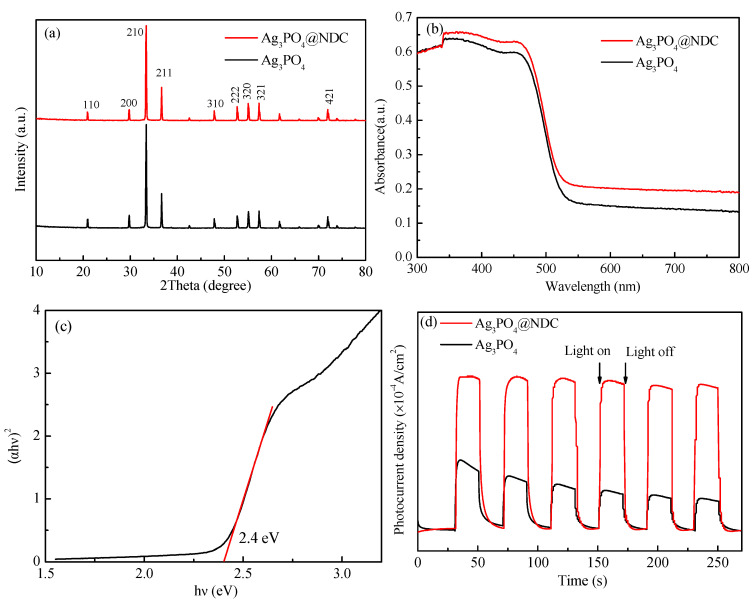
Ag_3_PO_4_ and Ag_3_PO_4_@NDC XRD spectrum (**a**), UV Vis DRS spectrum (**b**), fitting curve (**c**), photocurrent response curve (**d**).

**Figure 5 ijerph-19-14865-f005:**
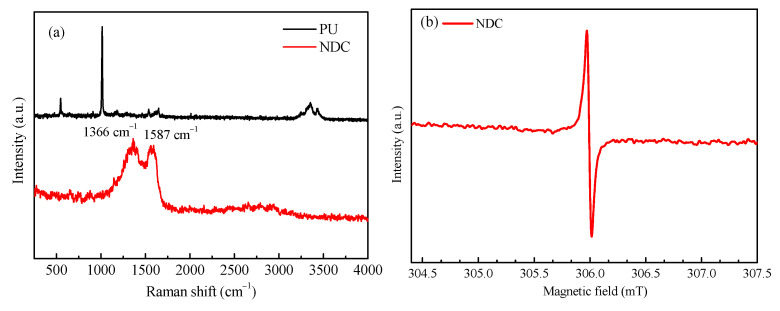
(**a**) Raman spectra of PU and NDC, (**b**) Low temperature EPR spectra of NDC.

**Figure 6 ijerph-19-14865-f006:**
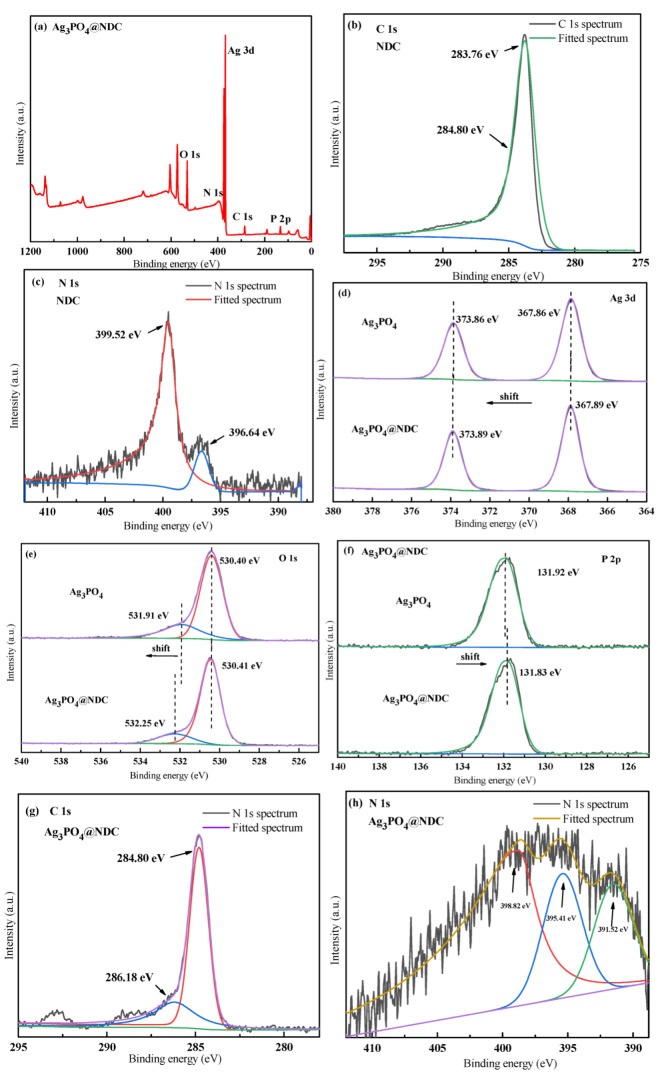
(**a**) XPS survey spectra of Ag_3_PO_4_@NDC; high-resolution XPS spectrum of NDC: (**b**) C 1s and (**c**) N 1s; high-resolution XPS spectrum of NDC and Ag_3_PO_4_@NDC: (**d**) Ag 3d, (**e**) O 1s, (**f**) P 2p; high-resolution XPS spectrum of Ag_3_PO_4_@NDC: (**g**) C 1s and (**h**) N 1s.

**Figure 7 ijerph-19-14865-f007:**
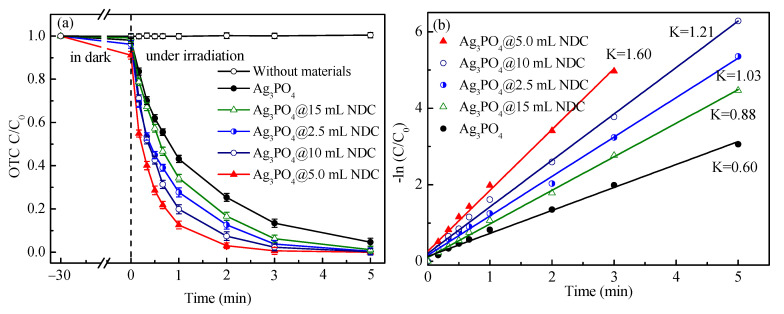
(**a**) Degradation curve of oxytetracycline by composite photocatalyst containing carbon materials with different proportions and silver phosphate monomer ([OTC]: 20 mg/L; catalysts: 50 mg; volume: 100 mL), (**b**) Pseudo first-order reaction kinetics curve fitted by degradation data.

**Figure 8 ijerph-19-14865-f008:**
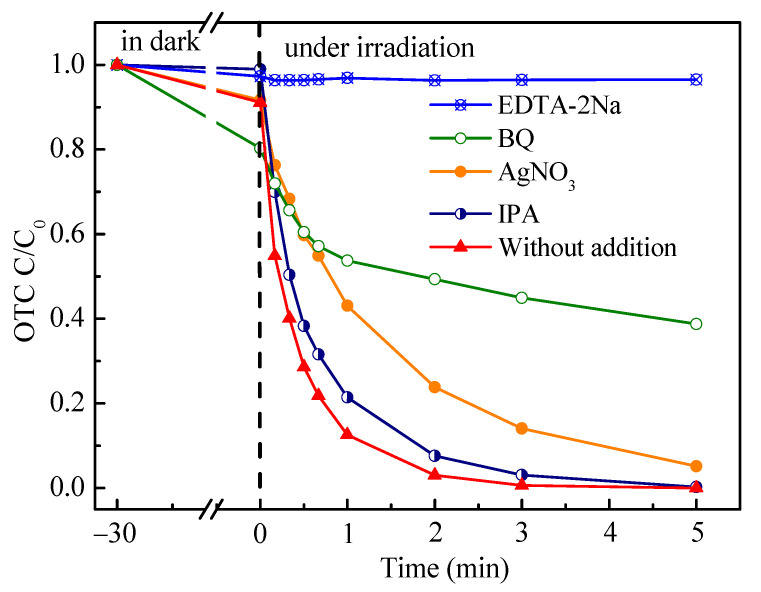
Degradation curve of oxytetracycline by Ag_3_PO_4_@5 mL NDC composite photocatalyst in the presence of different free radical traps.

**Figure 9 ijerph-19-14865-f009:**
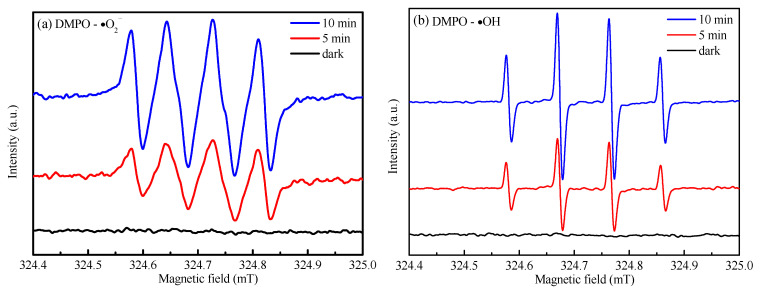
ESR spectra of Ag_3_PO_4_@5 mL NDC composite photocatalyst under dark and light conditions: (**a**) Capture of DMPO-·O_2_^−^ in methanol dispersion system; (**b**) DMPO-·OH capture in aqueous dispersion system.

**Figure 10 ijerph-19-14865-f010:**
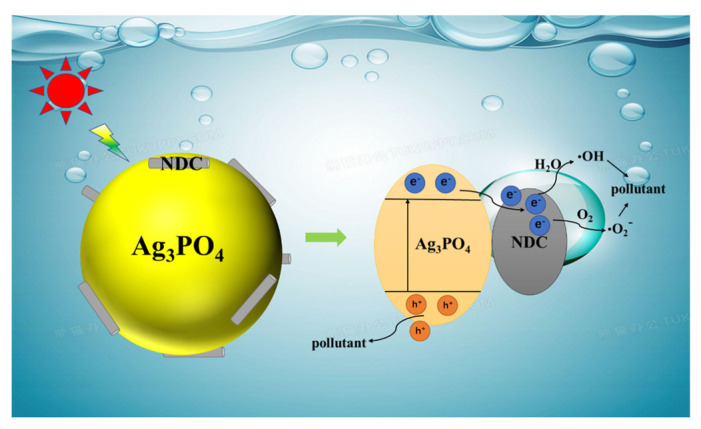
Degradation mechanism of Ag_3_PO_4_@NDC composite photocatalyst.

## Data Availability

Not applicable.

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
