# Peer review of "Highly Enhanced Photocatalytic Performances of Composites Consisting of Silver Phosphate and N-Doped Carbon Nanomesh for Oxytetracycline Degradation"

_ijerph, 2022, doi:10.3390/ijerph192214865_

Round 1

Reviewer 1 Report

This works present an interesting material design as a photocatalyst of oxytetracycline. The Authors describe systematically and nicely the background, ideas, methods and results. A few minor questions and suggestions are as follows,

1.  The tittle seems too long, could you shorten and elaborate the tittle?

2. In the experimental method section, please provide in more detail information regarding the parameter of each characterization technique and the name of each instrument.

3. There are Typos on Page 3 Line 119 (the solution the solution) and 120 (was was), please correct it accordingly

4. The caption for Figure 1F is missing

5.  Please describe in more detail information on the Figure caption such as Figure 2a-c, what is the difference and the main point for each sub figure

6. What is the chemical and physical mechanism that the composite with NDC can induce the ration of Ag3PO4 crystal lattice? Why after compositing the material, the (222) crystal face become more dominant?

7. Please put the unit in Y-axis of photocurrent density graph

8. Please check and correct the error and missing points in the caption of Figure 6.

Author Response

This works present an interesting material design as a photocatalyst of oxytetracycline. The Authors describe systematically and nicely the background, ideas, methods and results. A few minor questions and suggestions are as follows.

Author Response: Thank you very much for your comments.

1.The tittle seems too long, could you shorten and elaborate the tittle?

Author Response: Thank you for your kind suggeest. The tittle has been revised from “Synthesis of silver phosphate-based composite photocatalysts modified by nitrogen doped carbon nanomesh with defects structure and its degradation performance and mechanisms for oxytetracycline” to “Highly enhanced photocatalytic performance of silver phosphate for oxytetracycline degradation by combining with N-doped carbon nanomesh”.

2. In the experimental method section, please provide in more detail information regarding the parameter of each characterization technique and the name of each instrument.

Author Response: Thank you for your comment. The detail information regarding the parameter of each characterization technique and the name of each instrument have been provided in this manuscript.

3. There are Typos on Page 3 Line 119 (the solution the solution) and 120 (was was), please correct it accordingly.

Author Response: We are grateful for your kind correction, the extra words have been deleted in this revised manuscript.

4. The caption for Figure 1F is missing.

Author Response: Thank you for your comment. The caption of Figure 1f has been supplemented, which is also a SEM image of Ag3PO4@NDC.

5. Please describe in more detail information on the Figure caption such as Figure 2a-c, what is the difference and the main point for each sub figure.

Author Response: Thank you for your comment. The Figure 2a-c were the TEM images of NDC at different magnification. As shown in Figure 2a, NDC is a nanosheet with two-dimensional structure composed of nanotubes. From Figures 2b and 2c, it could be found that with the increase of magnification, there were uniform pore structures on the carbon material, which was due to the carbon defects caused by the removal of uniformly distributed urea units on the aromatic polymer chain of PU after annealing.

6. What is the chemical and physical mechanism that the composite with NDC can induce the ration of Ag3PO4 crystal lattice? Why after compositing the material, the (222) crystal face become more dominant?

Author Response: Thank you for your comment. The similar phenomenon has been reported, that is, the addition of carbon materials changes the exposure of silver phosphate crystal surface [1-2]. The results may be that there is electrostatic self-driven assembly between negatively charged NDC surface and positively charged Ag+, which will change the growth mode, size and morphology of silver phosphate crystal particles. Thus, the exposed crystal facets of silver phosphate changes. In addition, since the (222) crystal plane of silver phosphate has the highest surface energy, its exposure will be more concerned [3]. More in-depth chemical and physical mechanisms will be further investigated in the future research.

References:

[1] Lin, Y., Wu, S., Yang, C., Chen, M., Li, X. Preparation of size-controlled silver phosphate catalysts and their enhanced photocatalysis performance via synergetic effect with MWCNTs and PANI. Applied Catalysis B: Environmental 2019, 245, 71-86, doi: 10.1016/j.apcatb.2018.12.048.

[2] Lin, Y., Yang, C., Wu, S., Li, X., Chen, Y., Yang, W.L. Construction of Built‐In Electric Field within Silver Phosphate Photocatalyst for Enhanced Removal of Recalcitrant Organic Pollutants. Advanced Functional Materials 2020, 30(38), 2002918, doi: 10.1002/adfm.202002918.

[3] Bi, Y., Ouyang, S., Umezawa, N., Cao, J., Ye, J. Facet effect of single-crystalline Ag3PO4 sub-microcrystals on photocatalytic properties. Journal of the American Chemical Society 2011, 133(17), 6490-2, doi: 10.1021/ja2002132.

7. Please put the unit in Y-axis of photocurrent density graph.

Author Response: The unit in Y-axis of photocurrent density graph has been provided.

8. Please check and correct the error and missing points in the caption of Figure 6.

Author Response: We are grateful for your kind correction. The caption of Figure 6 has been revised into “Figure 6. (a) XPS survey spectra of Ag3PO4@NDC; high resolution XPS spectrum of NDC: (b) C 1s and (c) N 1s; high resolution XPS spectrum of NDC and Ag3PO4@NDC: (d) Ag 3d, (e) O 1s, (f) P 2p; high resolution XPS spectrum of Ag3PO4@NDC: (g) C 1s and (h) N 1s”.

Thanks a lot!

Reviewer 2 Report

This paper presents some results on the degradation of antibiotics by the conventional silver phosphate system modified by (NDC) nitrogen containing carbon materials.   This is an aspect to be investigated.   The paper deals with most of the aspects of the reaction studied and also characterized the catalyst preparations.

The following points to be noted:

1. The XPS results are sound enough but the assignments may have to be justified especially that of O1s spectrum above 530 eV?

2. The main reason for the observed activity of the system is the life time of the electron hole recombination rate is affected. This aspect needs some elaboration.

3. The microscopic examination is probably alright but the origin of these structures has to be identified.

Author Response

This paper presents some results on the degradation of antibiotics by the conventional silver phosphate system modified by (NDC) nitrogen containing carbon materials.   This is an aspect to be investigated.   The paper deals with most of the aspects of the reaction studied and also characterized the catalyst preparations.

The following points to be noted:

Author Response: Thank you very much for your comments.

1. The XPS results are sound enough but the assignments may have to be justified especially that of O1s spectrum above 530 eV?

Author Response: We are grateful for your kind correction. The wrong statement has been revised, the characteristic peaks at 530.4 eV and 531.9 eV were bulk lattice O of Ag3PO4 and surface O of catalysts, respectively.

2. The main reason for the observed activity of the system is the life time of the electron hole recombination rate is affected. This aspect needs some elaboration.

Author Response: Thank you for your comments. We agree with the comment that “The main reason for the observed activity of the system is the life time of the electron hole recombination rate is affected”. Because the transient photocurrent responses has been widely used to investigate the photogenerated charge separation of photocatalysts. In this study, the transient photocurrent response curve was also invetigated, it could be found that the Ag3PO4@NDC composite had much higher photocurrent density than that of pure Ag3PO4 under the same conditions. The results indicaed that the introduction of NDC significantly promoted the separation of photogenerated electron and hole in the photocatalytic reaction system of the composite catalyst, and produced more effective photogenerated carriers, which was favorable for achieving enhanced visible light photocatalytic activity.

3. The microscopic examination is probably alright but the origin of these structures has to be identified.

Author Response: Thank you for your comments. The SEM image of Figure 1a was the precursor PU. After annealing, the PU precursor formed NDC (Figure 1b and 1c). The pure Ag3PO4 as shown in Figure 1d, which had a polyhedral structure. After the addition of NDC, the morphologies of Ag3PO4@NDC composite were presented in the Figure 1e and Figure 1f.

Thanks a lot!

Reviewer 3 Report

Here, the authors prepared a photocatalyst material by the combination of nitrogen doped carbon (NDC) with carbon defects and Ag3PO4, named as Ag3PO4@NDC, which achieves 100% photocatalytic degradation efficiency of oxytetracycline (OTC). The improved performance was mainly mainly due to the intense interface interaction and electron transfer process at the interface between Ag3PO4 and NDC. This study provides a highly efficient visible light catalytic material, which can be a valuable reference for the design and development of new highly efficient visible light catalyst. Accordingly, I recommend this paper to be published after solving the attached issues.

1. In Figure 3, the origin morphology (SEM image) should be offered for better understanding the element distribution.

2. The caption of Figure 1f is missing, please add; sample mark inserted in Figure 6g is C 1s spectrum, but not N 1s spectrum; the whole paper should be carefully checked and solve the similar issues.
3. It’s recommended to provide a mini table to compare the performance between this material and others.

Author Response

Here, the authors prepared a photocatalyst material by the combination of nitrogen doped carbon (NDC) with carbon defects and Ag3PO4, named as Ag3PO4 which achieves 100% photocatalytic degradation efficiency of oxytetracycline (OTC). The improved performance was mainly mainly due to the intense interface interaction and electron transfer process at the interface between Ag3PO4 and NDC. This study provides a highly efficient visible light catalytic material, which can be a valuable reference for the design and development of new highly efficient visible light catalyst. Accordingly, I recommend this paper to be published after solving the attached issues.

Author Response: Thank you very much for your comments.

1. In Figure 3, the origin morphology (SEM image) should be offered for better understanding the element distribution.

Author Response: Thank you for your suggestion. The origin morphology (SEM image) has been provided in the revised supporting information for better understanding the element distribution.

2. The caption of Figure 1f is missing, please add; sample mark inserted in Figure 6g is C 1s spectrum, but not N 1s spectrum; the whole paper should be carefully checked and solve the similar issues.

Author Response: We are grateful for your kind correction.The caption of Figure 6 has been revised into “Figure 6. (a) XPS survey spectra of Ag3PO4@NDC; high resolution XPS spectrum of NDC: (b) C 1s and (c) N 1s; high resolution XPS spectrum of NDC and Ag3PO4@NDC: (d)Ag 3d, (e) O 1s, (f) P 2p; high resolution XPS spectrum of Ag3PO4@NDC: (g) C 1s and (h) N 1s”. In additon, the whole mauscript has been thoroughly checked to solve the similar issues.

3. It’s recommended to provide a mini table to compare the performance between this material and others.

Author Response: Thank you for your kind suggestion. The performance between this material and others has been summarized in the Table S1. Comprehensive consideration and comparison the apparent rate constant, removal rate and reaction time, it could be found that the photocatalytic performance of Ag3PO4@NDC prepared in this study was obviously better than that of previous reported other materials.

Table S1 Comparison of catalytic ability of Ag3PO4@NDC with other reported materials towards OTC removal

Photocatalysts

Pollutant

concentration (mg/L)

Catalysts

dosage (g/L)

Time (min)

Removal (%)

Rate  constants (min-1)

Reference

Ag3PO4@NDC

20

0.5

3

100

1.6

this study

CoFe@NSC

50

0.2

150

82.7%

0.00765

[4]

Nb2O5/g-C3N4

20

1.0

150

76.2%

0.01

[5]

BPTCN

10

1.0

60

81.05%

0.0276

[6]

OCN-24-550

20

1.0

120

85.76%

0.0164

[7]

Cu3P-ZnSnO3-g-C3N4

10

1.0

60

54.71%

-

[8]

FeOOH QDs/CQDs/g-C3N4

10

0.5

60

85.5%

0.0333

[9]

References

[4] Zhang, S.; Zhao, S.; Huang, S.; Hu, B.; Wang, M.; Zhang, Z.; He, L.; Du, M. Photocatalytic degradation of oxytetracycline under visible light by nanohybrids of CoFe alloy nanoparticles and nitrogen-/sulfur-codoped mesoporous carbon. Chemical Engineering Journal 2021, 420, 130516. doi: 10.1016/j.cej.2021.130516.

[5] Hong, Y.; Li, C.; Zhang, G.; Meng, Y.; Yin, B.; Zhao, Y.; Shi, W. Efficient and stable Nb2O5 modified g-C3N4 photocatalyst for removal of antibiotic pollutant. Chemical Engineering Journal 2016, 299, 74-84, doi: 10.1016/j.cej.2016.04.092.

[6] Wang, W.; Niu, Q.; Zeng, G.; Zhang, C.; Huang, D.; Shao, B.; Zhou, C.; Yang, Y.; Liu, Y.; Guo, H.; Xiong, W.; Lei, L.; Liu, S.; Yi, H.; Chen, S.; Tang, X. 1D porous tubular g-C3N4 capture black phosphorus quantum dots as 1D/0D metal-free photocatalysts for oxytetracycline hydrochloride degradation and hexavalent chromium reduction. Applied Catalysis B: Environmental 2020, 273, 119051, doi:10.1016/j.apcatb.2020.119051.

[7] Guo, H.; Niu, C.; Feng, C.; Liang, C.; Zhang, L.; Wen, X.-J.; Yang, Y.; Liu, H.; Li, L.; Lin, L. Steering exciton dissociation and charge migration in green synthetic oxygen-substituted ultrathin porous graphitic carbon nitride for boosted photocatalytic reactive oxygen species generation. Chemical Engineering Journal  2020, 385, 123919, doi: 10.1016/j.cej.2019.123919.

[8] Guo, F.; Huang, X.; Chen, Z.; Cao, L.; Cheng, X.; Chen, L.; Shi, W. Construction of Cu3P-ZnSnO3-g-C3N4 p-n-n heterojunction with multiple built-in electric fields for effectively boosting visible-light photocatalytic degradation of broad-spectrum antibiotics. Separation and Purification Technology 2021, 265, 118477, doi: 10.1016/j.seppur.2021.118477.

[9] Zhang, M.; Lai, C.; Li, B.; Xu, F.; Huang, D.; Liu, S.; Qin, L.; Fu, Y.; Liu, X.; Yi, H.; Zhang, Y.; He, J.; Chen, L. Unravelling the role of dual quantum dots cocatalyst in 0D/2D heterojunction photocatalyst for promoting photocatalytic organic pollutant degradation. Chemical Engineering Journal 2020, 396, 125343, doi: 10.1016/j.cej.2020.125343.

Thanks a lot!
